# A national dataset of 30-m annual urban extent dynamics (1985-2015) in the conterminous United States

Xuecao Li[1], Yuyu Zhou[*1], Zhengyuan Zhu[2], Wenting Cao[1]

[1] Department of Geological and Atmospheric Sciences, Iowa State University, Ames, IA, 50011, USA

[2] Department of Statistics, Iowa State University, Ames, IA 50011, USA

*Correspondence to*: Yuyu Zhou (yuyuzhou@iastate.edu)

**Abstract.** Dynamics of the urban extent at fine spatial and temporal resolutions over large areas are crucial for developing urban growth models and achieving sustainable development goals. However, there are limited practices of mapping urban

dynamics with these two merits combined. In this study, we proposed a new method to map urban dynamics from Landsat time series data using the Google Earth Engine (GEE) platform and developed a national dataset of annual urban extent (1985-2015) at a fine spatial resolution (30m) in the conterminous United States (US). First, we derived the change information of urbanized years in four periods that were determined from the National Land Cover Database (NLCD), using a temporal segmentation approach. Then, we classified urban extents in the beginning (1985) and ending (2015) years at the cluster level

through implementing a change vector analysis (CVA) based approach. We also developed a hierarchical strategy to apply the CVA based approach due to the spatially explicit urban sprawl over large areas. The overall accuracy of mapped urbanized years is around 90% with the one-year tolerance strategy. The mapped urbanized areas in the beginning and ending years are reliable, with overall accuracies of 96% and 88%, respectively. Our results reveal that the total urban area increased by about 20% during the period 1985-2015 in the US and the annual urban area growth is not linear over the years. Overall, the growth

pattern of urban extent in most coastal states is plateaued over the past three decades while the states in the Midwestern US show an accelerated growth pattern. The derived annual urban extents are of great use for relevant urban studies such as urban area projection and urban sprawl modelling over large areas. Moreover, the proposed mapping framework is transferable for developing annual dynamics of urban extent in other regions and even globally. The data are available at https://doi.org/10.6084/m9.figshare.8190920.v2 (Li et al., 2019c).

## 1 Introduction

The rapid global urbanization causes environmental, ecological, and public concerns for human-beings for sustainable development goals (SDGs) (Rodriguez et al., 2018). Globally, urban area, commonly defined as the space dominated by the built environment (e.g., buildings, roads, and runways) from remote sensing, only accounts for a tiny fraction of the Earth's

surface (Schneider et al., 2010), however, it is home to the most global economy, population, energy consumption, and greenhouse gas emissions (Solecki et al., 2013). According to the latest *World Urbanization Prospects* (United Nations, 2019), more than 50% of the world's population lives in urban areas, and this percentage will increase to 66% by the middle of this century. Moreover, most urban population growth would likely to occur in developing regions, where the realization of SDGs faces more challenges because of potential risks from thermal environment change caused by urban heat island (Peng et al.,

2012), degradation of urban ecosystem services (Li et al., 2017;Irwin and Bockstael, 2007), energy consumption with changed environment and human activities (Güneralp et al., 2017;Zhou et al., 2014;Alberti et al., 2017), and public health concerns (Gong et al., 2012;Luber et al., 2014). Therefore, understanding the pathway of urban sprawl (i.e., expansion of the geographic extent of urban area) and developing advanced urban growth models are highly needed for adapting and mitigating potential risks under future urbanization (Li and Gong, 2016a;Weng, 2012).

The datasets of urban extent dynamics at the fine spatial (e.g., 30m) and temporal (e.g., annual) resolutions are the key to capture the rate, trend, and stage of urbanization for a better understanding of this process (Zhang et al., 2014). Such datasets can provide fine information about urban form (e.g., layout, geometry, and distribution), which can be further used for relevant studies such as urban energy consumption (Chen et al., 2011), biodiversity in urban ecosystem (Andersson and Colding, 2014), and air pollutant emissions (Fan et al., 2018). In addition, the relationship between urban dynamics and annual socioeconomic

development (e.g., population and gross domestic product) can help to better understand the reasons behind urbanization (Seto et al., 2002;Xie and Weng, 2017). Finally, the long temporal span (e.g., decades) of urban dynamics can capture a relatively complete process of urban sprawl with different stages (Li et al., 2019a;Gong et al., 2019;Cao et al., 2020;Gong et al., 2020). The information of long-term urban dynamics is valuable in developing urban growth models, such as investigating the generation and propagation of errors (or uncertainties) in urban spatial sprawl models (Santé et al., 2010;Li and Gong, 2016a).

However, current mapping approaches that focus on multi-temporal (e.g., decade and half-decade) urban extent are limited to

reflect the process of urban sprawl (e.g., acceleration or deceleration) of cities and explain their differences caused by demographic and socioeconomic drivers (Sexton et al., 2013).

Urban extent mapping at fine spatial and temporal resolutions, especially over large areas, is still lacking, although urban extent maps with a variety of spatial and temporal resolutions have been developed. For example, there are several global urban extent products such as that from the nighttime light data (1km) (Zhou et al., 2015;Zhou et al., 2018;Xie and Weng, 2016), the Moderate Resolution Imaging Spectroradiometer (MODIS) data (500m) (Schneider et al., 2010), and even the fine-resolution Landsat data (30m) (Chen et al., 2015;Gong et al., 2013;Liu et al., 2018). However, these existing multi-temporal national or global urban extent maps were generally built separately in each period, with limited consideration of the temporal consistency of urban growth (Li et al., 2015;Song et al., 2016;Shi et al., 2017).

There are several challenges in mapping urban extent at fine spatial and temporal resolutions over large areas. First, land use and cover changes in urban domains are complicated, with the inter-class conversions and multi-phase changes before urbanization or during the post-urbanization period (Li and Gong, 2016b;Lu and Weng, 2004). For example, various land cover types such as vegetation, water, and barren can be potentially converted to built-up areas, and such conversion may experience multiple phases, e.g., from highly vegetated land to low vegetated or barren, and then eventually to built-up areas with post-urbanization changes. Second, durations of land surface change introduced by urbanization are different across regions, that is, urban sprawl may occur within a short period or last for a couple of years in different regions (Song et al., 2016;Kennedy et al., 2010).

In general, two approaches have been used to derive spatiotemporally consistent urban extent maps from high spatial and temporal satellite observations. One is improving the classified urban time series using post-processing techniques (Liu and Cai, 2012;Li et al., 2015); the other one is identifying the change information using the continuous time series data of relevant indicators such as vegetation index (Huang et al., 2010;Kennedy et al., 2010). The first method requires intensive labor on collecting training samples for classification and specific post-processing techniques (Gong et al., 2013;Chen et al., 2015;Liu and Cai, 2012;Gong et al., 2019), which is challenging and time consuming for regional and global mapping over a long temporal span. The second one poses a new challenge for managing, manipulating, and analysing the massive amount of time series data over large areas.

Due to these challenges in mapping urban dynamics at fine spatial and temporal resolutions over large areas, it is in highly demand to develop a generalized and efficient mapping approach. In this study, we mapped the annual dynamics (1985-2015) of urban extent in conterminous United States (US) by developing a generalized and efficient mapping approach on the state-of-art Google Earth Engine (GEE) platform. The remainder of this paper describes the study area and data (Section 2), the proposed national-mapping approach (Section 3), the results with discussion (Section 4), the data availability (Section 5), and concluding remarks (Section 6).

## 2 Datasets

Landsat time series data on the GEE platform, spanning from 1985-2015, are the primary data source for mapping annual urban extent in this study. The advent of GEE is designed for planetary-scale studies using different sources of satellite images (Gorelick et al., 2017;Li et al., 2019b), and it is a good choice for mapping project over large areas. In this study, we used multiple L1T-level Landsat surface reflectance products, including the Thematic Mapper (TM), the Enhanced Thematic Mapper Plus (ETM+), and the Operational Land Imager (OLI). These products have been corrected for the radiometric, topographic, and atmospheric effects (Masek et al., 2006). All clean-sky pixels were used to composite the time series data for analyses, with clouds and their shadows removed. In total, around 460,000 Landsat scenes were used for the conterminous US over past three decades.

The national land cover database (NLCD) and nighttime light data are ancillary datasets in this study. The NLCD provides multi-temporal urban maps in 1992, 2001, 2006, and 2011 (Homer et al., 2015;Xian et al., 2009), which were used as the reference urban areas in these years. The NLCD has been widely used for its reliable performance at the national scale (Wickham et al., 2017;Wickham et al., 2010). In this study, we derived the urban extent map in 1992 using the NLCD 1992/2001 retrofit land cover change product, so that all urban extents derived from the NLCD in different years (i.e., 1992, 2001, 2006, and 2011) are comparable (Fry et al., 2009). Besides, nighttime light images of the Visible Infrared Imaging Radiometer Suite (VIIRS) were used to delineate the potential urban cluster after 2011 (Li and Zhou, 2017).

## 3 Method

In this study, we developed a new framework with a unique hierarchical strategy for mapping annual urban extents in large areas on the GEE platform using long-term Landsat observations (Fig. 1). First, we grouped the study period (1985-2015) into four periods, namely B1 (1985-1992), B2 (1992-2001), B3 (2001-2011), and F1 (2011-2015), based on the available NLCD.

For Landsat time series data in each period, we detected the urbanized years at the pixel level by implementing a temporal segmentation approach (Li et al., 2018) (Fig. 1a). Second, given that NLCD only provides urban extent from B2 to B3, we classified urbanized areas at the cluster level in the periods of B1 and F1 using a change vector analysis (CVA) based approach. We developed a hierarchical strategy to implement the CVA based approach due to the spatially explicit urban sprawl over large areas. That is, the CVA based approach was applied in potential urban clusters (derived from VIIRS data) in each grid

(around 250 km × 250 km), according to the size of potential urban clusters (Fig. 1b). Details of each procedure are presented in the following sections.

### 3.1 Detection of urbanized years

We preprocessed the raw Landsat time series data before implementing the temporal segmentation approach. We systematically corrected the OLI surface reflectance data to make it consistent with other sensors (i.e., TM and EMT+) as

suggested by Roy et al. (2016). After that, we generated the normalized difference vegetation index (NDVI), the modified normalized difference water index (MNDWI), and the shortwave infrared (SWIR) reflectance. These three indexes can well represent vegetation, water, and bare lands, respectively, and are primary conversion sources to urbanized areas (Li and Gong, 2016b). The annual maximal NDVI was used to represent the growth of vegetation because the NDVI has a distinctive seasonal pattern and the greenest season varies over different biomes, e.g., January-March in the western US and June-August in the

central US. The annual mean values of MNDWI and SWIR from all observations except for the winter time were used to composite the annual time series data.

We implemented the temporal segmentation approach for each urbanized pixel in four periods (i.e., B1, B2, B3, and F1). These urbanized pixels during each period were identified using urban extent maps derived from NLCD and classified results in B1 and F1 using the CVA based approach. Within each period, we identified the starting (P1) and ending (P2) years of change

using the temporal segmentation approach, according to the overall trend of the indicators (i.e., NDVI, MNDWI, and SWIR).

For urbanization from vegetation, the indicator of NDVI shows a decreasing trend (Fig. 2a), while curves of MNDWI and SWIR show increasing trends (Fig. 2b). In this temporal segmentation method, we first applied a linear regression to the annual time series data of three indicators (i.e., NDVI, MNDWI, and SWIR), and then determined these two turning points (i.e., P1 and P2) according to their annual residuals to the regression-based trend line. If the overall trend is decreasing, the years with the largest residuals above (below) and below (above) the regression-based trend line were identified as P1 and P2 (Fig. 2). The change year derived from the indicator with the largest change magnitude (i.e., change between P1 and P2) was identified as the final result. In addition, the duration of change is the difference of years between P1 and P2. More details about the temporal segmentation can be found in Li et al. (2018).

### 3.2 Classification of urbanized areas before 1992 and after 2011

We classified urbanized areas in periods of B1 (1985-1992) and F1 (2011-2015) using a CVA based approach at the national level (Fig. 1b). Urbanized areas of two middle periods (i.e., B2 and B3) were directly obtained from NLCD. Results from the temporal segmentation approach (i.e., change magnitude within each period) were used to identify urbanized areas in the CVA based approach in the beginning (B1) and ending (F1) periods. Full time series data were used in our CVA based approach, which is different from the commonly used approach based on a pair of images in two periods (Xian et al., 2009;Yu et al., 2016). The change magnitude ($\Delta V$) was calculated using three indicators (Eq. 1). Compared to the six spectral band information of Landsat, the three indicators show similar or even better performance in capturing the change magnitude (Fig, S1), as well as providing the information of conversion sources of urbanized areas. Pixels with a large $\Delta V$ were regarded as potentially changed areas. We identified these potentially changed areas using a multi-threshold approach because different conversions have different thresholds of $\Delta V$ (Eq. 2).

$$\Delta V = \sqrt{(NDVI_{t1} - NDVI_{t2})^2 + (MNDWI_{t1} - MNDWI_{t2})^2 + (SWIR_{t1} - SWIR_{t2})^2} \qquad (1)$$

$$CV_j = \begin{cases} 1, \Delta V_j \geq \mu_j + \alpha\sigma_j \\ 0, \Delta V_j < \mu_j + \alpha\sigma_j \end{cases} \qquad (2)$$

where $CV_j$ is the status of change (i.e., 1: change and 0: no change) for cover type $j$; $\mu_j$ and $\sigma_j$ are the mean and standard deviation of $\Delta V_j$; $t1$ and $t2$ are turning years of P1 (before change) and P2 (after change), respectively; and $\alpha$ is an adjustable parameter that was set as 1.5 in this study as suggested by Morisette and Khorram (2000).

We implemented the CVA based approach within urban masks in the first (B1) and last (F1) periods. For B1, the urban extent of NLCD 1992 was used as a potential urban mask before 1992. For the period of F1, an approximate urban extent derived from VIIRS data in 2015 (Li et al., 2018) was used as a potential urban boundary for classification. Within the derived urban boundary, we classified urban areas in 2015 using urban pixels sampled from NLCD 2011. Finally, we derived urbanized areas

in the period of F1, using the potential change areas from the CVA approach, the urban boundary from NTL data, and the urban extent from NLCD 2011. Pseudo changes that are not relevant to the urban sprawl were removed during this process. More details about the CVA based approach can be found in Li et al. (2018).

### 3.3 A hierarchical strategy on the GEE

We developed a hierarchical strategy to implement the CVA based approach at the national level. This strategy enables us to

detect urbanized areas over large areas with spatially explicit patterns of urban sprawl. In this strategy, we grouped all potential urban clusters into two categories using a size filter of 100 km$^2$ (Fig. S2) for implementing different thresholds in the CVA based approach to derive urbanized areas (Fig. 3). For those large clusters (i.e., larger than 100 km$^2$), we isolated each of them as an independent spatial unit and applied the CVA based approach on them. For the remaining small urban clusters within the same grid, we treated them as an integrated unit to derive urbanized areas.

### 4 Results and discussion

### 4.1 Annual urban growth

The annual growth of urban areas varies across years in the conterminous US, which cannot be revealed by the NLCD (Fig. 4). Overall, the average growth rate at the national scale is around 1,000 km$^2$/y during 1985-2015. The total increment is about 31,000 km$^2$, around 20% relative to the urban area in 1985 (Fig. 4a). Our results provide more details of urban dynamics

according to the annual growth rate (km$^2$/y) of urban areas, compared to the growth rate (km$^2$/y) of the NLCD in each period (Fig. 4b). The mean growth rates of NLCD are 1,015 km$^2$/y, 1,512 km$^2$/y, and 929 km$^2$/y, during the periods of 1992-2001, 2001-2006, and 2006-2011, respectively. However, the annual dynamics within each period are notably different. In general, there are notably decreasing trends of growth during periods of 1997-2001 and 2007-2010, and a profound increasing trend

during 2004-2006. Particularly, the decreasing trend during 2006-2011 is the most significant, with a total decrease from 1,380 km$^2$/y in 2007 to 520 km$^2$/y in 2010, which is likely caused by the financial crisis around 2008.

The annual growth of urban areas is different across states. There is an overall increasing trend at the early years and a decreasing trend at the latter years in period of 2001-2011 (Fig. 5). The mean growth rate in all states is 25 km$^2$/y. Texas (TX), Florida (FL), and California (CA) are three states with the highest growth rates, which are 117 km$^2$/y, 93km$^2$/y, and 80 km$^2$/y, respectively. In general, in most states, their relative changes of annual growth in urban area are higher than the mean growth rate of NLCD at the early years. After that, their relative changes of annual growth is below the mean growth rate. This trend is consistent with NLCD results, with a declined mean growth rate around 40% during period 2006-2011, relative to the 2001-2006 (Fig. 4b). It is worthy to note that NLCD in 2006 was not used in our mapping approach. The comparison of the urban area growth during 2001-2006 shows a good agreement between our results and NLCD (Fig. S3). Therefore, the NLCD in 2006 independently indicates that our approach can well capture the dynamic of urban areas.

A distinctive urban area growth was observed for cities with a rapid population growth. We chose the top 10 cities in the US based on the population growth rate during 2010-2017 (Fig. 6). Most of them are in the Southern and the Eastern US, such as TX, FL, and North Carolina (NC). Overall, the growth of urban areas in these top 10 cities is significant. The rank-based urban area growth agrees well with the result from population growth (Fig. S4). For example, there is a remarkable urban sprawl around 2006-2015 in the Village city (FL), which is also the city with the fastest population growth among the top 10.

## 4.2 Long-term patterns of urban growth

Our annual urban extent data reveal different long-term patterns of urban growth across states in the US during the past three decades. We calculated the percentage of urban area growth relative to the base year 1985 for each state from 1986 to 2015. States show different patterns (i.e., convex or concave hull) according to the time series of relative change (Fig. 7a). As such, we defined two urban area growth patterns by comparing the derived time series curve to the reference line (Fig. 7b). If the curve of urban area growth is overall above the reference line (e.g., Missouri (MO)), then we regarded this pattern as a plateaued growth; on the contrary, it belongs to an accelerated growth (e.g., Arizona (AZ)). In general, urban area growth patterns in most coastal states are plateaued, and states in the South and the Midwestern US show an accelerated growth pattern in general (Fig. 7c). In particular, the relatively accelerated growth of urban areas over past three decades in agricultural states

such as Iowa (IA), North Dakota (ND), and South Dakota (SD) challenges the sustainable development of agriculture system. Also, the annual urban areas over a long term indicate the urban area growth is not linear over the years, although the linear growth of urban areas was widely used in urban sprawl modelling, if only the coarse-temporal resolution urban extent data are available (Li et al., 2014;Sexton et al., 2013).

## 4.3 Conversion sources of urbanized areas

The primary conversion sources of urbanized areas are different across states and change over time (Fig. 8). Most urbanized areas were converted from cropland and forest, within a relatively short duration (i.e., 1~3 years) (Fig. S5-6). Overall, vegetation (i.e., cropland, forest, grass, and shrub) is the dominated source of urbanized areas over all the states and years. In particular, the cropland is the most predominant source of urbanized areas, accounting for 46% of the total urbanized areas during 1992-2015. Besides, there is a certain percentage of urbanized areas converted from water or wetland in some states in the Eastern and Southern coastal areas, e.g., FL, Louisiana (LA), and South Carolina (SC). Additionally, percentages of land cover encroached by urban vary over the years. For example, the percentage of encroached cropland decreases, while the encroached grass increases in North Dakota (ND).

## 4.4 Evaluation

### 4.4.1 Detected urbanized years

The identified urbanized years using the temporal segmentation approach agree well with the manually interpreted result using samples from NLCD, with an overall accuracy of around 90% using the one-year tolerance strategy (Fig. 9). We visually interpreted more than 500 samples that randomly collected from urbanized regions from NLCD during periods of B1, B2, and B3 (Fig. S7), aided by multi-temporal Landsat images, Google Earth high resolution images, and time series data of relevant indicators (i.e., NDVI, MNDWI, and SWIR) (Li et al., 2018). Period of F1 is not included due to its short-term (2011-2015). Given that there are uncertainties in the manual interpretation, we validated our results using the identified absolute year and the one-year tolerance strategy (Song et al., 2016). The overall accuracies of B1, B2, and B3 without the one-year tolerance strategy are 58%, 48%, and 57%, respectively. When the one-year tolerance strategy was used, their agreements were considerably improved to 89%, 83%, and 88%, respectively (Fig. 9). The adoption of the one-year strategy is reasonable

because the urban sprawl may occur in the beginning or ending phases of a given year, which may cause confusions among neighbouring years (Song et al., 2016;Huang et al., 2010).

The spatial pattern of detected urbanized years is reliable through the visual inspection in eight selected representative cities, with different urban sprawl rates during 1985-2015 and population sizes ranging from 200,000 to 900,000 (Fig. 10). In general, urban areas in these cities expanded from the center to the fringe areas, whereas the pathways of urban sprawl are notably different among these cities. For example, the direction of urban sprawl is opposite between Des Moines (IA) and Memphis (TN). The snapshots in Fig. 11 suggest a good agreement of urbanized years between our results and Landsat observations. For example, most urbanized areas in Las Vegas (Region B in Fig. 11) occurred after 2000, which is consistent with our mapped urbanized years (i.e., pixels colored from yellow to red). Similar cases can also be found in other regions such as Des Moines (A) and Kansas (C) (Fig. 11).

### 4.4.2 Classification of urbanized areas in periods of B1 and F1

The CVA based approach performs well for classifying urbanized areas, according to the accuracy assessment using samples randomly generated on both non-urban and urbanized areas during periods of B1 (1985-1992) and F1 (2011-2015) (Table 1). Validation samples for period B1 were randomly collected from persistent urban areas since 1985 and urbanized areas during 1985-1992. For the period of F1, samples were generated based on non-urban areas and urbanized areas during 2011-2015, within the VIIRS derived potential urban boundary (Fig. S8). The manual interpretation is based on the time series of Landsat and high-resolution Google Earth images in the two periods. The overall accuracies of classified urbanized areas for periods of B1 and F1 are 96% and 88%, respectively (Table 1). The higher accuracy in the period B1 compared with the period of F1 is because the validation samples in this period are within the possible urban extent of 1992 from the NLCD. Also, the misclassified urbanized areas in the period of F1 are mainly caused by the confusion between bared land (e.g., rocks, or dry soil) and urban with similar spectral features (Mertes et al., 2015).

### 4.4.3 Uncertainties of annual urban extent data

There are several sources of uncertainties in our annual urban extent data. The first is the classification error in the NLCD, despite this is the most reliable database in the US with a fine resolution and multiple periods (Homer et al., 2015). On the one

hand, the detected change information is incorrect in the misclassified urbanized pixels from NLCD. On the other hand, for those urbanized pixels but not identified in NLCD, their change information is not captured in our result. However, the overall accuracy of land cover classification in NLCD is about 85%~90% (Wickham et al., 2017), and the accuracy of urban land cover is even higher (i.e., larger than 95% in selected examples of US) (Li et al., 2018). Moreover, the CVA based approach

can be implemented to improve the urban extent maps of NLCD as change magnitudes of those pseudo urban pixels in the NLCD are notably lower than changes caused by urban sprawl. In addition, the omitted urbanized pixels in the NLCD can be potentially captured using the CVA based approach. The second is the classification error in mapped urbanized areas at the beginning and ending years. Uncertainties caused by spectral similarities between urban and bared lands could still exist in our results (Table 1), although we have used different constraints (e.g., change vector, classification results, and NTL) to

mitigate such uncertainties. More advanced classification algorithms and additional information such as thermal features could be helpful for improving our algorithm in monitoring urban dynamics.

**5 Data availability**

The generated data of annual urban dynamics are available at https://doi.org/10.6084/m9.figshare.8190920.v2 (Li et al., 2019c). The dataset is organized by state (total 49) in the conterminous US. Location of US states can be found in the figure

of "US_State.jpg". Full names and abbreviations of US states are provided in the file of "US_StateList.xls". The data are in GeoTIFF with the georeferenced information embedded. Each file was projected to the Albers Equal Area Conic projection, with a spatial resolution of 30m. The legend of GeoTIFF file can be founded in the figure of "Legend.jpg". The lookup table between pixel values (1-31) and urbanized years (1985-2015) can be found in the file of "Year_Code_Loopup.csv". The national land cover database was collected from the U.S. Geological Survey at https://www.mrlc.gov/, and the VIIRS nighttime

light data were downloaded from the National Oceanic and Atmospheric Administration at https://ngdc.noaa.gov/.

**6 Conclusions**

In this study, we mapped annual urban extents in the conterminous US by developing an efficient framework on the GEE platform using long-term Landsat observations. First, aided by the NLCD, we temporally grouped the entire temporal span

into four periods (i.e., B1: 1985-1992, B2: 1992-2001, B3: 2001-2011, and F1: 2011-2015). Then, we derived the urbanized years and change magnitudes measured by indicators of NDVI, MNDWI, and SWIR at the pixel level, using a temporal segmentation approach in each period. After that, we classified urbanized areas at the cluster level in the beginning (1985) and the ending (2015) years, through implementing a CVA based approach. Considering the spatially explicit urban sprawl over large areas, we developed a unique hierarchical strategy to apply the CVA based approach at the national level. Finally, the mapped urban dynamics in these four periods were combined as a complete dataset of 30-year dynamics of urban extent in the conterminous US.

The proposed mapping framework with the unique hierarchical strategy achieves a good performance in mapping annual dynamics of urban extent at a fine spatial resolution at the national level. The overall accuracies of detected urbanized years for periods of B1, B2, and B3 are 89%, 83%, and 88%, respectively, with a one-year tolerance strategy. Meanwhile, the CVA based approach on the output from temporal segmentation can classify urbanized areas well, with over accuracies of 96% and 88% for periods of B1 and F1, respectively. Also, the implementation of CVA based approach using the proposed hierarchical strategy can capture the heterogeneity of urban growth over different regions, periods, and urban sizes, which helps to build a reliable dataset of urban dynamics.

There is a notable difference in growth rates and patterns of annual urban area across states in the US over the past three decades. The total increment of urban areas is about 31,000 km$^2$, which accounts for around 20% of the urban area in 1985. The long-term growth of urban areas is not linear over the years. The results suggest there is an increasing trend of urban area growth in the early years of 2001-2011 and then a decreasing trend in the latter years. Using the annual time series data of urban areas, we observed a plateaued growth pattern of urban areas in most coastal states and an accelerated growth pattern in the Midwestern US. Besides, the cropland is the most predominant source of increased urban areas, accounting for 46% of the total urbanized areas during 1992-2015.

This study provides a successful application of mapping annual urban extent at the national scale, through combining existing good-quality NLCD urban extent maps, long-term Landsat time series data, and the GEE cloud-based platform. The proposed approach can be transferred to other regions with similar multi-temporal land cover datasets as the NLCD, for updating existing land cover datasets with a higher temporal resolution. This study opens a new avenue to use all available Landsat observations

for mapping annual urban extent at the national level compared with previous studies using the supervised classification or post-processing (Schneider et al., 2010;Li et al., 2015;Liu and Cai, 2012). Moreover, the derived change information from the temporal segmentation using annual observations is more reliable compared with the research using a pair of Landsat images in two years (Yu et al., 2016). However, this approach may introduce uncertainties if the composited annual time series Landsat observations fluctuate too much, especially when this fluctuation is larger than the change induced by urbanization.

**Author contributions**

ZY and LX designed the research; LX and ZY implemented the research and wrote the paper; ZZ and CW revised the manuscript.

**Competing interests**

The authors declare that they have no conflict of interest.

**Acknowledgments**

This research was funded in part by cooperative agreement 68-7482-17-009 between the USDA Natural Resources Conservation Service and Iowa State University. We would like to thank organizations that shared their datasets for use in this study.

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

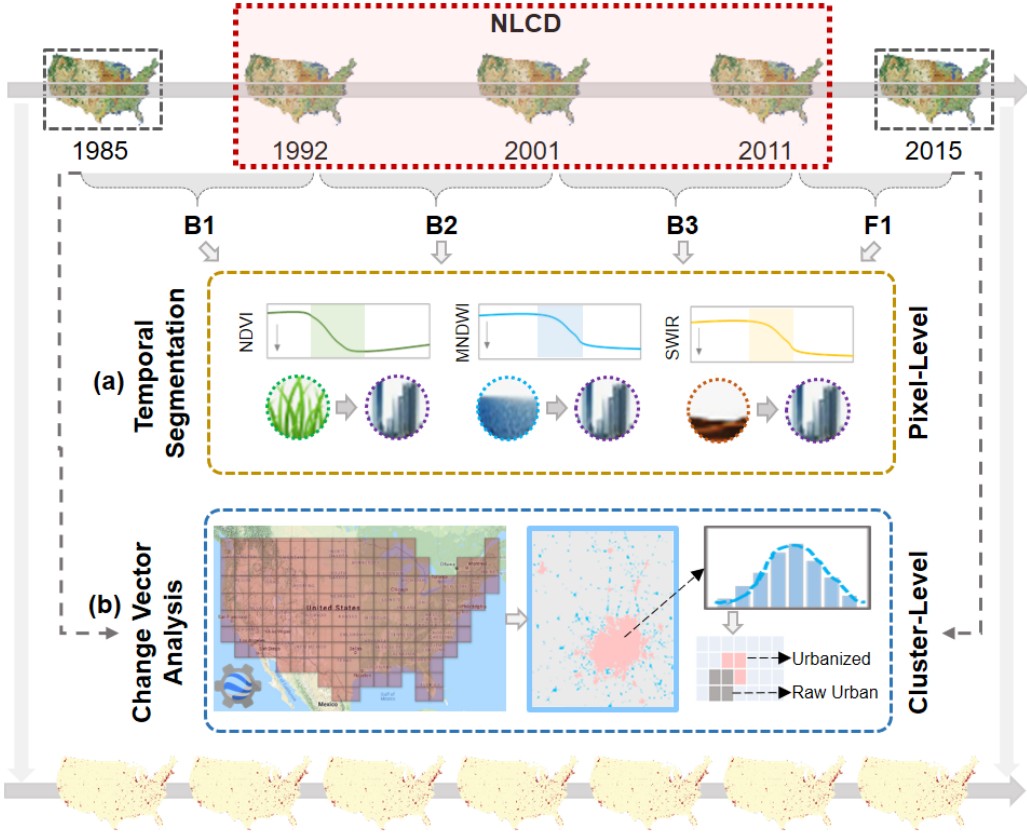

**Fig. 1:** The proposed framework for mapping annual urban extent dynamics in the conterminous US through detecting the urbanized year at the pixel level using temporal segmentation approach (a) and classifying urbanized areas at the cluster level in periods of B1 and F1 using change vector analysis (b). Basemap data ©2019 Google Inc.

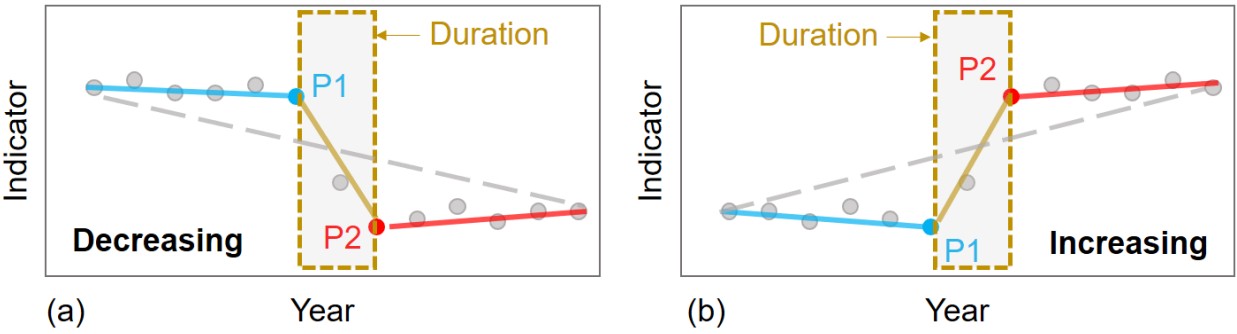

**Fig. 2:** Illustration of the temporal segmentation approach using indicators with decreasing (NDVI) (a) or increasing (MNDWI and SWIR) (b) trends.

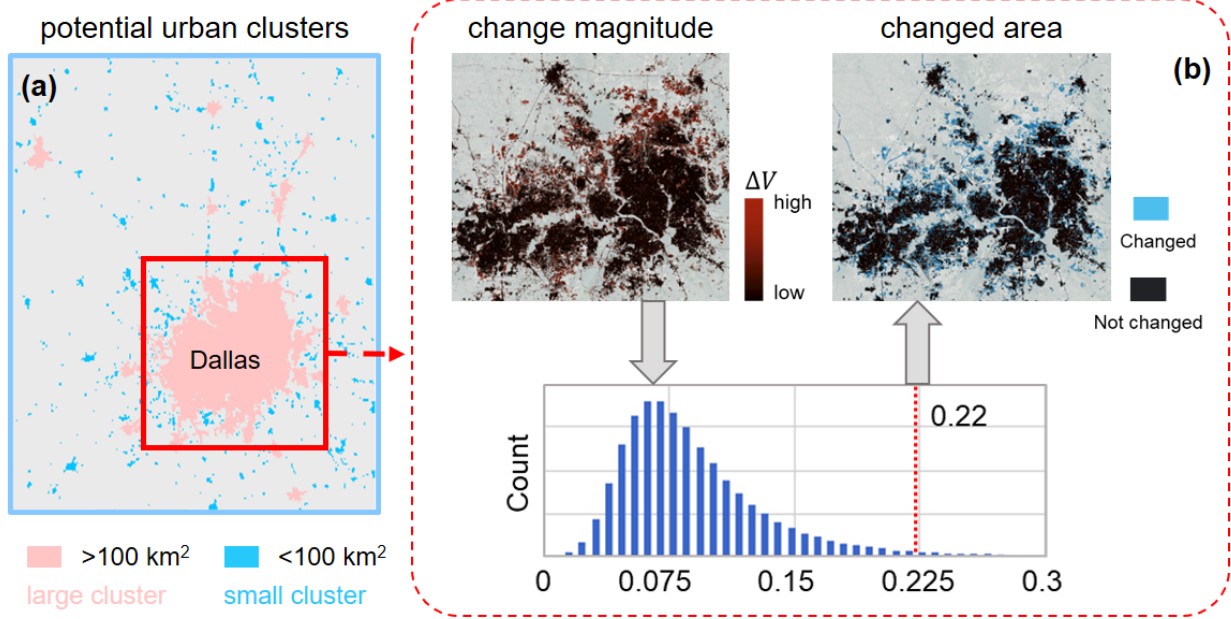

**Fig. 3:** An illustration of the CVA based approach. An example grid with potential urban clusters including Dallas, Texas for implementing the CVA based approach (a). An example of the CVA based approach in the cluster of Dallas, Texas (b). The change magnitude is the difference of three indicators (i.e., NDVI, MNDWI, and SWIR) before and after the urbanized year, and the changed areas are pixels with magnitudes greater than the determined threshold ($\mu + 1.5\sigma$) from the histogram (dotted red line). $\mu$ and $\sigma$ are the mean and standard derivation of change magnitudes in the potential urban cluster.

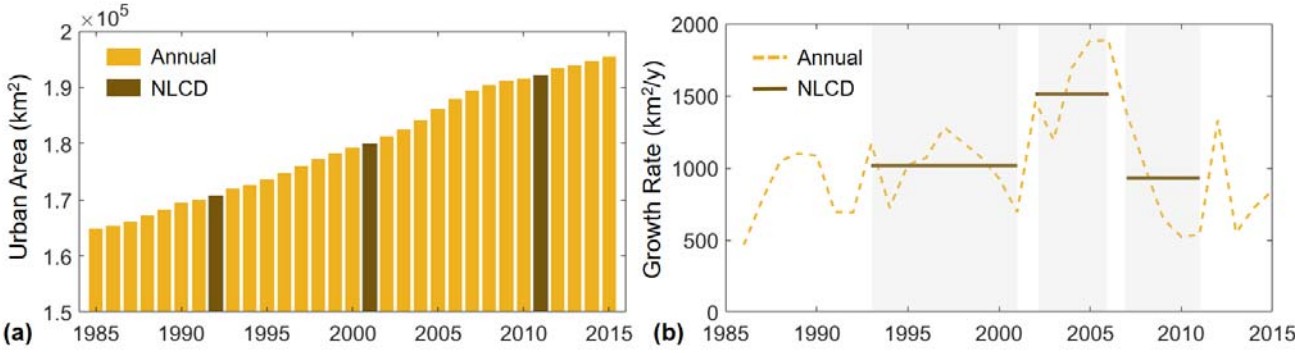

**Fig. 4:** Annual growth of urban areas in the conterminous US (1985-2015) (a) and their annual growth rates (km²/y) compared to the NLCD in three periods (shadow frames) of 1992-2001, 2001-2006, and 2006-2011(b).

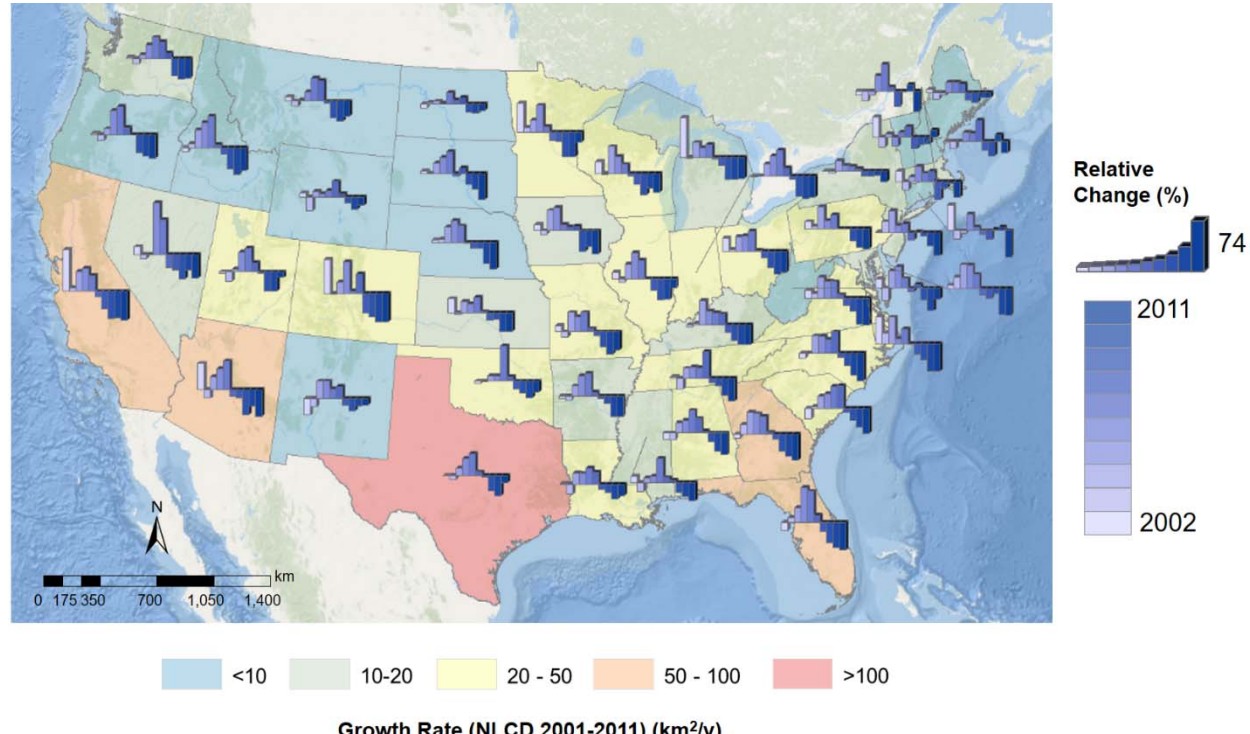

**Fig. 5:** State-based relative change of annual growth of urban areas compared to the mean growth rate of NLCD during period 2001-2011. Basemap data ©2019 Esri Inc.

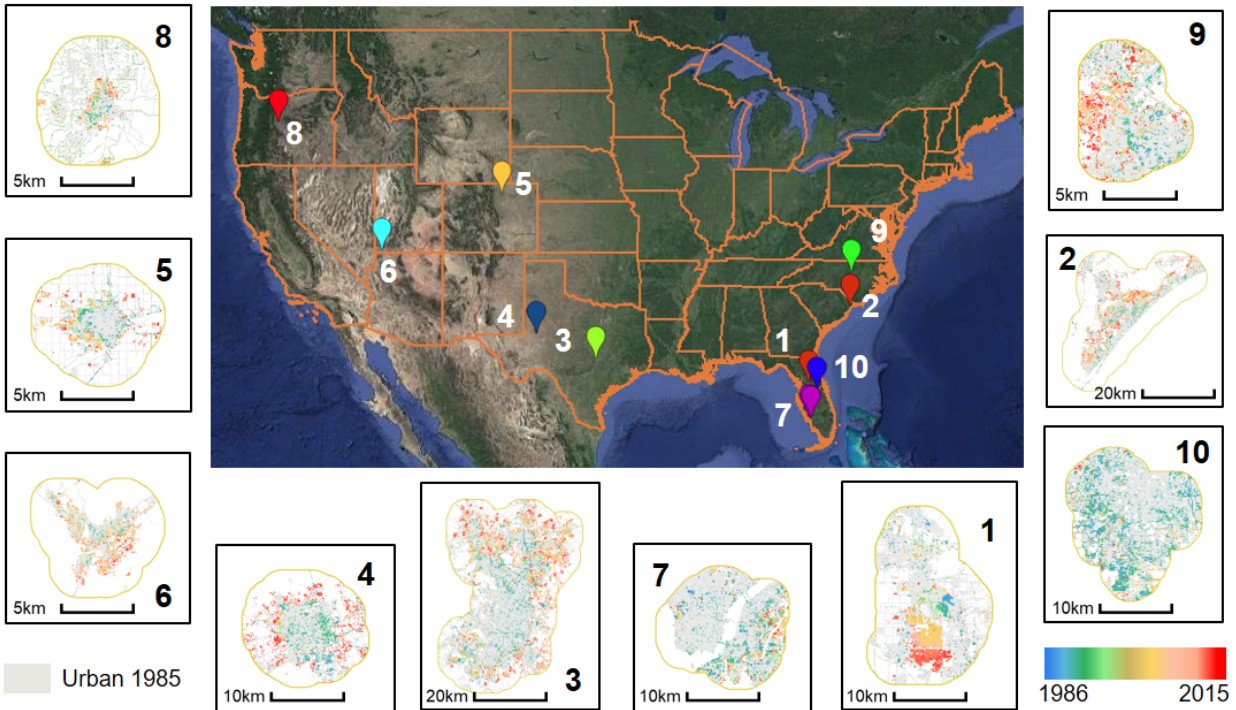

**Fig. 6:** An illustration of urban area growth in the top 10 fast-growing cities in the US according to the population growth during 2010-2017.

1: Village (Florida), 2: Myrtle Beach (South Carolina / North Carolina), 3: Round Rock (Texas), 4: Midland (Texas), 5: Greeley (Colorado),

6: St. George (Utah), 7: Fort Myers (Florida), 8: Redmond (Oregon); 9: Raleigh (North Carolina), 10: Orlando (Florida). Basemap data

©2019 Google Inc.

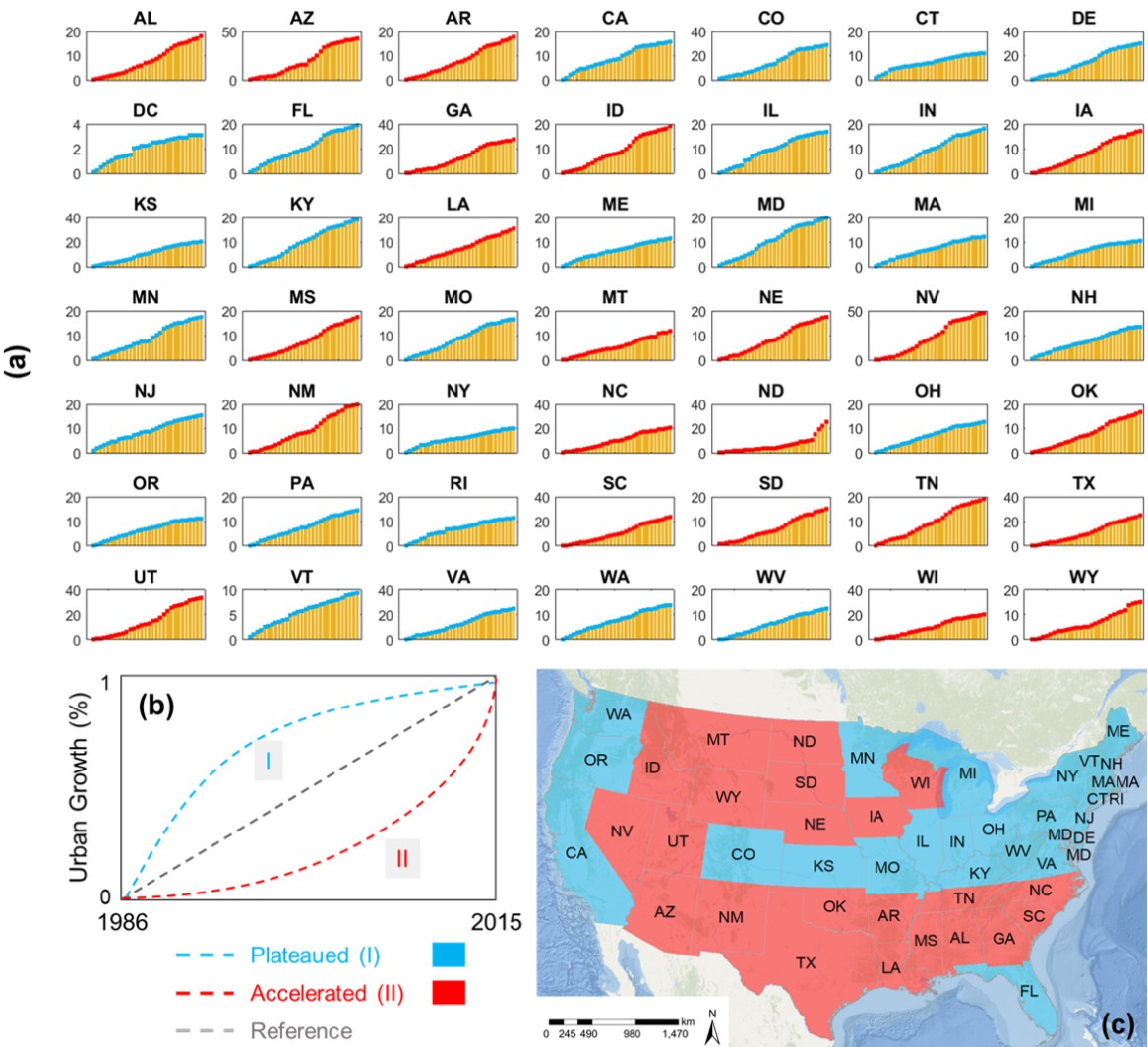

**Fig. 7:** Urban area growth patterns over past three decades of each state in the US (a), the proposed conceptual model (b), and the classified urban area growth types (c). Basemap data ©2019 Esri Inc.

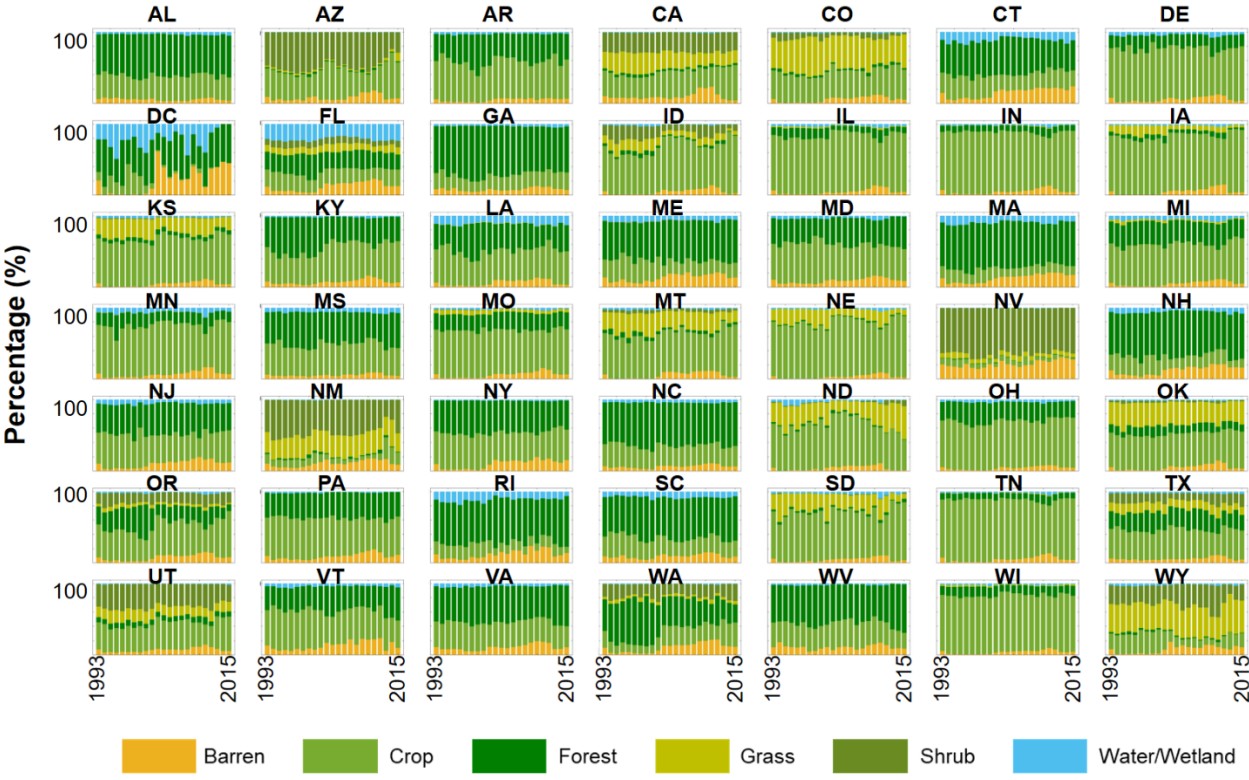

**Fig. 8:** Conversion sources of urbanized areas during 1992-2015.

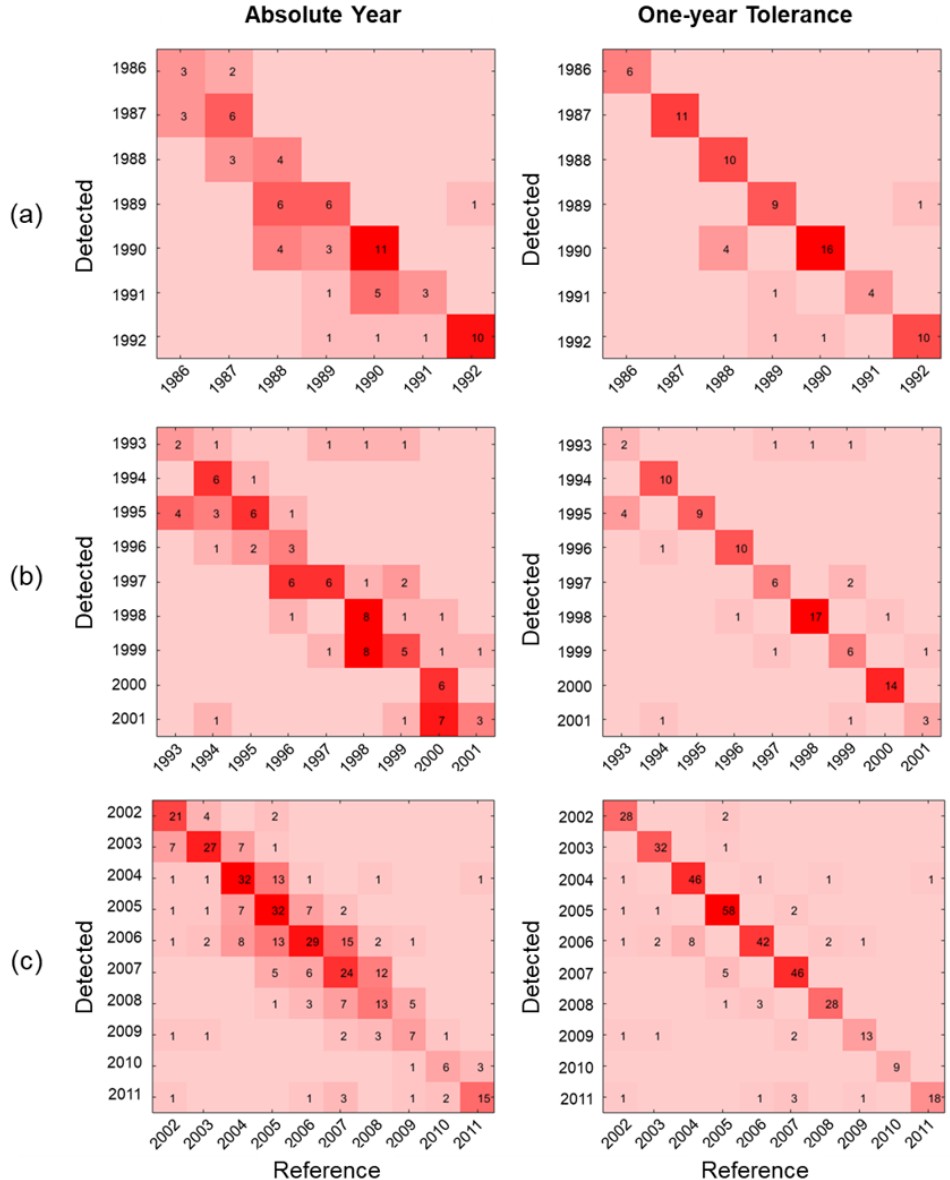

**Fig. 9:** Accuracy assessment of urbanized years over different periods of B1 (1985-1992) (a), B2 (1992-2001) (b), and F1 (2001-2011) (c).

Each grid labels the number of urbanized year from the manual interpretation and our approach.

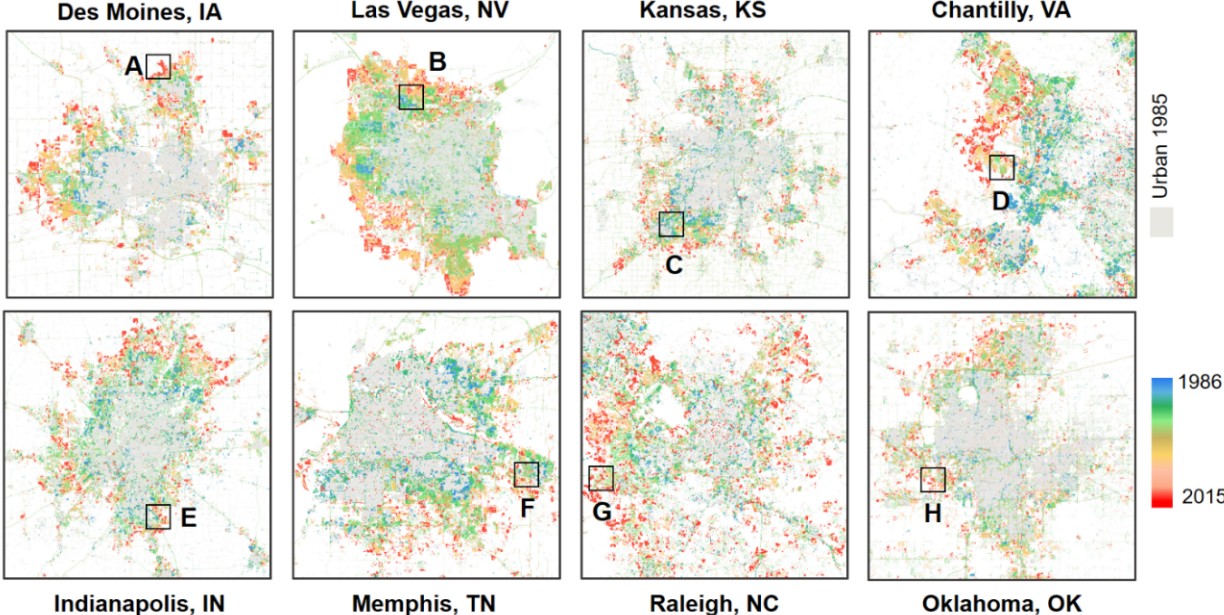

**Fig. 10:** Annual dynamics of urban extent in eight selected US cities over past three decades. The black frames are regions in Fig. 11 for further comparison with Landsat images.

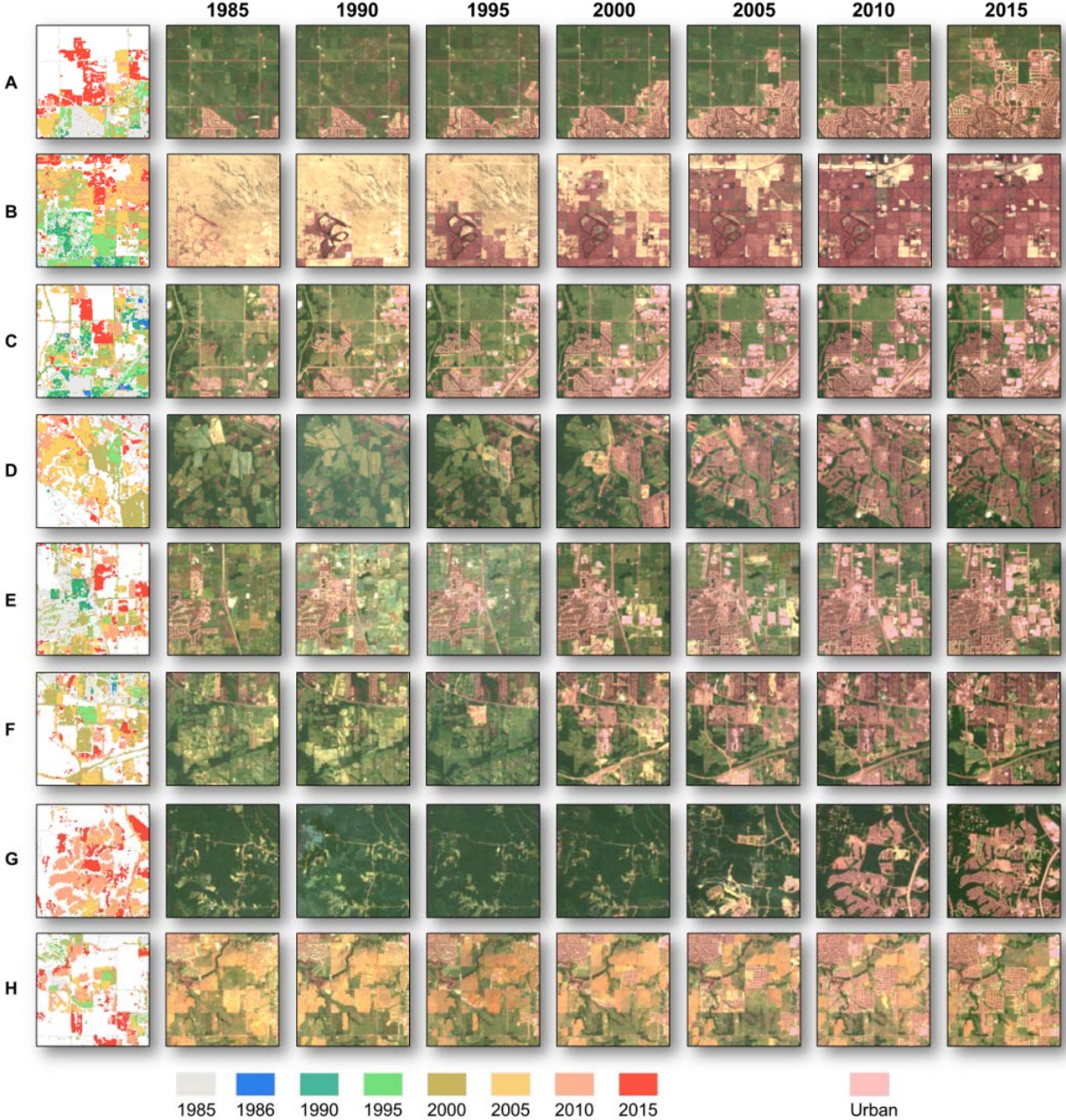

**Fig. 11:** Comparison of annual urban dynamics with Landsat images. The geographic location of each region (A-H) corresponds to the black frames in Fig. 10. The spatial extent of each region is 25 km$^2$.

**Table 1.** Accuracy assessment of classified urbanized areas for periods of B1 and F1.

| | **Reference** | | | |
|---|---|---|---|---|
| | **1985-1992** | No-Change | Change | Producer's accuracy (%) |
| **Mapped** | No-Change | 99 | 1 | 99% |
| | Change | 8 | 92 | 92% |
| | User's accuracy (%) | 93% | 99% | |
| | Overall accuracy | 96% | Kappa | 0.91 |
| | **2011-2015** | No-Change | Change | Producer's accuracy (%) |
| **Mapped** | No-Change | 92 | 8 | 92% |
| | Change | 17 | 83 | 83% |
| | User's accuracy (%) | 84% | 91% | |
| | Overall accuracy | 88% | Kappa | 0.75 |