# Peer review of "A national dataset of 30-m annual urban extent dynamics (1985-2015) in the conterminous United States"

_Earth System Science Data, 2019_

## Referee Comment (RC1) · Anonymous Referee #1 · 26 Aug 2019

The manuscript outlined a research result that used Landsat historic datasets and Google Earth Engine to develop a 30 m annual urban extent in the United States. The mapped urban extents reached an overall accuracy between 96% and 88%. In general, these accuracy levels for urban mapping, especially for mapping the change in such long term and large scale, are very impressive. The result also shows high agreement with the existing national land cover database. The manuscript is also well prepared.

However, I have following major comments for the manuscript.

1. The research used change vector as the foundational change detection tool to

characterize urban change as an annual base. The authors did not use conventional change vector approach, which uses all spectral band information. Only three derived indexes were used to build up the change vector. The author did not explain why these three indexes were used. Are they best choice? No matter what answer is, sensitivity tests are necessary to compare with other approaches using other indexes or full spectral bad information.

2. Fig. 4. These are interesting graphics. However, colors for these lines (Fig.4b) makes these graphics hard to read. Should use different colors to clearly illustrate annual growth rate.

3. Fig11. The colors of Landsat images are confused. It is hard to compare your mapped urban extents with satellite images. More clear and meaningful graphics are needed to clearly illustrate urban extent change and corresponding images.

---

## Referee Comment (RC2) · Anonymous Referee #2 · 25 Sep 2019

This study provides a long-term nation-wide annual urban extent data at the 30m spatial resolution, with high accuracy. Such fine temporal and spatial dataset is highly desirable, and is potentially very useful for the academia community, as well for practitioners. Additionally, the proposed approach also has the potential to be adopted in other regions to obtain similar dataset. With that being said, I also have some concerns about the current version of the manuscript, and think it can be improved by considering the comments below. 1) This is a very important dataset that would be interesting to people in many fields. The authors may want to expand their introduction section to include more discussions about how this dataset can be applied. 2) It would be helpful to define some of the key terms in the manuscript. Examples are urban and urban

sprawl. This is because these terms are often vaguely defined, but different disciplines may define them differently. 3) The temporal segmentation approach is very interesting and important. While more details can be found in Li et al. (2018), it would be helpful to include more details in this manuscript 4) The authors used the four time slices NLCD data as a baseline. But the NLCD data themselves have errors in classification. I did not mean it's not OK to use these datasets. But how such errors may affect the annual classification results? Are there better ways to address the issue of existing errors in these datasets? The authors may want to address this issue in discussion. 5) The accuracy assessment. The authors selected samples for accuracy assessment differently for the time periods of 1992-2011 and that of 1985-1992 and 2011-2015. The authors visually interpreted more than 500 samples that randomly collected from urbanized regions from NLCD during periods of B1, B2, and B3, but for the period of 2011-2015, samples were generated based on both non-urban areas and urbanized areas during. Why use two different approaches? How might using samples only from urbanized regions from NLCD affect the results of accuracy assessment? 6) How the approach can be further refined to obtain even better results? Some discussion on future research would be helpful.

---

## Author Comment (AC1) · 23 Oct 2019

Reviewer #1: The manuscript outlined a research result that used Landsat historic datasets and Google Earth Engine to develop a 30 m annual urban extent in the United States. The mapped urban extents reached an overall accuracy between 96% and 88%. In general, these accuracy levels for urban mapping, especially for mapping the change in such long term and large scale, are very impressive. The result also shows high agreement with the existing national land cover database. The manuscript is also well prepared. However, I have following major comments for the manuscript.

Response: thank you very much for your positive comments. Below please find pointby-point responses.

**1-1: The research used change vector as the foundational change detection tool to characterize urban change as an annual base. The authors did not use conventional change vector approach, which uses all spectral band information. Only three derived indexes were used to build up the change vector. The author did not explain why these three indexes were used. Are they best choice? No matter what answer is, sensitivity tests are necessary to compare with other approaches using other indexes or full spectral bad information.**

Response: thank you for your suggestion. We explained the reason of using three indicators (i.e., NDVI, MNDWI, and SWIR) in our change vector analysis (CVA) approach, and compared the performance of the CVA approach using three indicators and six spectral bands as suggested. First, three indicators of NDVI, MNDWI, and SWIR can well represent vegetation, water, and bare lands, respectively. In the urban environment, these three land covers are primary conversion sources to urbanized areas. We clarified it in our revised manuscript.

"After that, we generated the normalized difference vegetation index (NDVI), the modified normalized difference water index (MNDWI), and the shortwave infrared (SWIR) reflectance. These three indexes can well represent vegetation, water, and bare lands, respectively, and are primary conversion sources to urbanized areas (Li and Gong, 2016a)."(page 5, line 15-18).

Second, we compared the performance of the CVA approach using three indicators and six spectral bands as suggested (Fig. S1) in the Chicago region in the period of 2001-2011. We found the derived $\Delta V1$ from three indicators performs similarly with or even better than $\Delta V2$ from six spectral bands (Fig. S1, a-b, d-e, and h-j). The overall accuracy of the derived potential urbanized map from $\Delta V1$ is better than that from $\Delta V2$ (Fig. S1, c and f). We clarified this in the revised manuscript.

"The change magnitude ($\Delta V$) was calculated using three indicators (Eq. 1). Compared

to the six spectral band information of Landsat, the three indicators show similar or even better performance in capturing the change magnitude (Fig, S1), as well as providing the information of conversion sources of urbanized areas." (page 6, line 15-17).

Insert Fig. S1

Fig. S1. Comparison of the performance of the change vector analysis (CVA) approach using three indicators (i.e., NDVI, MNDWI, and SWIR) and six spectral bands (i.e., B1, B2, B3, B4, B5, and B7) in the Chicago region during 2001-2011. Change vectors of $\Delta V1$ (a) and $\Delta V2$ (d) and their histograms (b and e) were derived from three indicators and six spectral bands, respectively. The detected change areas from $\Delta V1$ and $\Delta V2$ are presented in (c) and (f), respectively, and they were further compared with the reference data of NLCD (g). Enlarged examples are given in (h), (i), and (j), respectively. The dotted line in histograms (b and e) are determined thresholds.

**1-2: Fig. 4. These are interesting graphics. However, colors for these lines (Fig.4b) makes these graphics hard to read. Should use different colors to clearly illustrate annual growth rate.**

Response: thank you for your suggestion. We revised Fig. 4 using two different colors to illustrate annual growth rates of NLCD and our results as below.

Insert Fig. 4

Fig. 4: Annual growth of urban areas in the conterminous US (1985-2015) (a) and their annual growth rates (km2/y) compared to the NLCD in three periods (shadow frames) of 1992-2001, 2001-2006, and 2006-2011(b).

**1-3: Fig11. The colors of Landsat images are confused. It is hard to compare your mapped urban extents with satellite images. More clear and meaningful graphics are needed to clearly illustrate urban extent change and corresponding images.**

Response: as suggested we improved our figure for better illustration. We improved the color scheme of urban dynamics maps to show a better comparison between urban

extent changes with corresponding Landsat images (Fig. 11).

Insert Fig. 11

Fig. 11: Comparison of annual urban dynamics with Landsat images. The geographic location of each region (A-H) corresponds to the black frames in Fig. 10. The spatial extent of each region is 25 km2.

Please also note the supplement to this comment:
https://www.earth-syst-sci-data-discuss.net/essd-2019-107/essd-2019-107-AC1-supplement.pdf

<hr style="width:40%;margin:0">

(a) **ΔV1** 0.95 0

(b) ×10⁵ Count 2 1 0 0.1270 μ=0.0423 σ=0.0565 0 0.05 0.1 0.15 0.2 **ΔV1**

(c) OA: 0.82

(h)

(i)

(j)

Change (NLCD)

(g)

(d) **ΔV2** 1.54 0

(e) ×10⁴ 15 Count 10 5 0 0.0823 μ=0.0264 σ=0.0372 0 0.05 0.1 0.15 0.2 **ΔV2**

(f) OA: 0.73 Urban

**Fig. 1.** Fig.S1

[Figure]

**Fig. 2.** Fig.4

[Figure]

**Fig. 3.** Fig.11

---

## Author Comment (AC2) · 23 Oct 2019

Reviewer #2: This study provides a long-term nation-wide annual urban extent data at the 30m spatial resolution, with high accuracy. Such fine temporal and spatial dataset is highly desirable, and is potentially very useful for the academia community, as well for practitioners. Additionally, the proposed approach also has the potential to be adopted in other regions to obtain similar dataset. With that being said, I also have some concerns about the current version of the manuscript, and think it can be improved by considering the comments below.

Response: thank you very much for your positive comments. Below please find point-

[Figure]

by-point responses.

**2-1: This is a very important dataset that would be interesting to people in many fields. The authors may want to expand their introduction section to include more discussions about how this dataset can be applied.**

Response: thank you for your suggestion. We improved the Introduction section with added discussion about applications of our dataset.

"The datasets of urban extent dynamics at the fine spatial (e.g., 30m) and temporal (e.g., annual) resolutions are the key to capture the rate, trend, and stage of urbanization for a better understanding of this process (Zhang et al., 2014). Such datasets can provide fine information about urban form (e.g., layout, geometry, and distribution), which can be further used for relevant studies such as urban energy consumption (Chen et al., 2011), biodiversity in urban ecosystem (Andersson and Colding, 2014), and air pollutant emissions (Fan et al., 2018). In addition, the relationship between urban dynamics and annual socioeconomic development (e.g., population and gross domestic product) can help to better understand the reasons behind urbanization (Seto et al., 2002; Xie and Weng, 2017). Finally, the long temporal span (e.g., decades) of urban dynamics can capture a relatively complete process of urban sprawl with different stages (Li et al., 2019a). The information of long-term urban dynamics is valuable in developing urban growth models, such as investigating the generation and propagation of errors (or uncertainties) in urban spatial sprawl models (Santé et al., 2010;Li and Gong, 2016b)." (page 2, line 21-24).

**2-2: It would be helpful to define some of the key terms in the manuscript. Examples are urban and urban sprawl. This is because these terms are often vaguely defined, but different disciplines may define them differently.**

Response: thank you for your suggestion. We defined these terms as suggested.

"Globally, urban area, commonly defined as the space dominated by the built environment (e.g., buildings, roads, and runways) from remote sensing, only accounts for a tiny fraction of the Earth's surface (Schneider et al., 2010)" (page 2, line 3-5).

"Therefore, understanding the pathway of urban sprawl (i.e., expansion of the geographic extent of urban area) and developing advanced urban growth models are highly needed for adapting and mitigating potential risks under future urbanization (Li and Gong, 2016b;Weng, 2012)." (page 2, line 12-14).

**2-3: The temporal segmentation approach is very interesting and important. While more details can be found in Li et al. (2018), it would be helpful to include more details in this manuscript.**

Response: thank you for your suggestion. We added more details of the temporal segmentation approach.

"We implemented the temporal segmentation for each urbanized pixel in four periods (i.e., B1, B2, B3, and F1). These urbanized pixels during each period were identified using urban extent maps derived from NLCD and classified results in B1 and F1 using the CVA based approach. Within each period, we identified the starting (P1) and ending (P2) years of change using the temporal segmentation approach, according to the overall trend of the indicators (i.e., NDVI, MNDWI, and SWIR). For urbanization from vegetation, the indicator of NDVI shows a decreasing trend (Fig. 2a), while curves of MNDWI and SWIR show increasing trends (Fig. 2b). In this temporal segmentation method, we first applied a linear regression to the annual time series data of three indicators (i.e., NDVI, MNDWI, and SWIR), and then determined these two turning points (i.e., P1 and P2) according to their annual residuals to the regression-based trend line. If the overall trend is decreasing, the years with the largest residuals above (below) and below (above) the regression-based trend line were identified as P1 and P2 (Fig. 2). The change year derived from the indicator with the largest change magnitude (i.e., change between P1 and P2) was identified as the final result. In addition, the duration of change is the difference of years between P1 and P2." (page 5, line 22-page 6, line

7).

**2-4: The authors used the four time slices NLCD data as a baseline. But the NLCD data themselves have errors in classification. I did not mean it's not OK to use these datasets. But how such errors may affect the annual classification results? Are there better ways to address the issue of existing errors in these datasets? The authors may want to address this issue in discussion.**

Response: thank you for your suggestion. We discussed uncertainties of our annual urban extent results due to NLCD and potential approaches to mitigate this influence in our revised manuscript. We added a new section of 4.4.3 Uncertainties of annual urban extent data.

"There are several sources of uncertainties in our annual urban extent data. The first is the classification error in the NLCD, despite this is the most reliable database in the US with a fine resolution and multiple periods (Homer et al., 2015). On the one hand, the detected change information is incorrect in the misclassified urbanized pixels from NLCD. On the other hand, for those urbanized pixels but not identified in NLCD, their change information is not captured in our result. However, the overall accuracy of land cover classification in NLCD is about 85%~90% (Wickham et al., 2017), and the accuracy of urban land cover is even higher (larger than 95% in selected examples of US) (Li et al., 2018). Moreover, the CVA based approach can be implemented to improve the urban extent maps of NLCD as change magnitudes of those pseudo urban pixels in the NLCD are notably lower than changes caused by urban sprawls. In addition, the omitted urbanized pixels in the NLCD can be potentially captured using the CVA based approach." (page 10, line 23-page 11, line 7).

**2-5: The accuracy assessment. The authors selected samples for accuracy assessment differently for the time periods of 1992-2011 and that of 1985-1992 and 2011-2015. The authors visually interpreted more than 500 samples that randomly collected from urbanized regions from NLCD during periods of B1, B2, and B3, but for the period**

of 2011-2015, samples were generated based on both non-urban areas and urbanized areas during. Why use two different approaches? How might using samples only from urbanized regions from NLCD affect the results of accuracy assessment?

Response: thank you for your questions. We clarified the use of two validation strategies in different periods. As we described in the method section, the timing (i.e., year) of urbanized pixels was identified using the temporal segmentation approach, while the urbanized areas in the beginning (1985) and ending (2015) years (not available in NLCD) were classified using the modified change vector analysis approach. Accordingly, we used two different validation strategies: one is for the accuracy of identified urbanized years, and another is for the accuracy of classified urbanized areas. Our evaluation using urbanized samples from NLCD during three periods (i.e., B1, B2, and B3) was used for identified urbanized years from the temporal segmentation approach. For periods of B1 (1985-1992) and F1 (2011-2015), we assessed the accuracy of classified urbanized areas in 1985 and 2015, respectively, using samples in both non-urban and urbanized areas. We clarified these issues in our revised manuscript.

"The identified urbanized years using the temporal segmentation approach agree well with the manually interpreted result using samples from NLCD, with an overall accuracy of around 90% using the one-year tolerance strategy (Fig. 9)." (page 9, line 16-17).

"The CVA based approach performs well for classifying urbanized areas, according to the accuracy assessment using samples randomly generated on both non-urban and urbanized areas during periods of B1 (1985-1992) and F1 (2011-2015)(Table 1)." (page 10, line 12-13).

**2-6: How the approach can be further refined to obtain even better results? Some discussion on future research would be helpful.**

Response: thank you for your suggestion. As suggested, we discussed potential improvements in future research. First, errors of the reference urban extent maps can be mitigated, and we discussed this in our response to comment #2-4. Second, there are

uncertainties caused by spectral similarities between urban and bare lands and these uncertainties can be mitigated with the help of other data when they become available.

"The second is the classification error in mapped urbanized areas at the beginning and ending years. Uncertainties caused by spectral similarities between urban and bared lands could still exist in our results (Table 1), although we have used different constraints (e.g., change vector, classification results, and NTL) to mitigate such uncertainties. More advanced classification algorithms and additional information such as thermal features could be helpful for improving our algorithm in monitoring urban dynamics." (page 11, line 7-11).

Please also note the supplement to this comment:
https://www.earth-syst-sci-data-discuss.net/essd-2019-107/essd-2019-107-AC2-supplement.pdf